# Influence of chitosan and chitosan oligosaccharide on dual antibiotic-loaded bone cement: *In vitro* evaluations

**Saran Tantavisut**[1,2]**, Jiraporn Leanpolchareanchai**[3]**, Amaraporn Wongrakpanich**[3]*****

**1** Department of Orthopaedic, Chulalongkorn University, Pathumwan, Bangkok, Thailand, **2** Hip Fracture Research Unit, Chulalongkorn University, Pathumwan, Bangkok, Thailand, **3** Department of Pharmacy, Faculty of Pharmacy, Mahidol University, Ratchathewi, Bangkok, Thailand

\* amaraporn.won@mahidol.ac.th

**Data Availability Statement:** All relevant data are within the paper and its supporting Information files.

## Abstract

### Background and purpose

The purpose of this study was to investigate the effect of incorporating chitosan (Ch) and chitosan oligosaccharides (ChO) into the commercially premixed antibiotic-loaded bone cement (ALBC). We compare antibiotic release profiles, antibacterial activity, and mechanical properties among different ALBC formulations. The hypothesis was that increasing the amount of Ch and ChO in the cement mixture would increase the antibiotics released and bacterial control. ALBC mixed with Ch or ChO may create a greater effect due to its superior dissolving property.

### Materials and methods

The bone cement samples used in this project were made from Copal® G+V composed of vancomycin and gentamicin. To prepare the Ch and the ChO mixed bone cement samples, different amounts of Ch and ChO were added to the polymethylmethacrylate matrix with three concentrations (1%, 5%, and 10%). Drug elution assay, antimicrobial assay, *in vitro* cytotoxicity, and mechanical properties were conducted.

### Results

Bone cement samples made from Copal® G+V alone or combined with Ch or ChO can release vancomycin and gentamicin into the phosphate-buffered saline. Mixing ChO into the bone cements can increase the amount of drug released more than Ch. ChO 10% gave the highest amount of antibiotics released. All samples showed good antibacterial properties with good biocompatibility *in vitro*. The microhardness values of the Ch and ChO groups increased significantly compared to the control group. In all groups tested, the microhardness of bone cements was reduced after the drug eluted out. However, this reduction of the Ch and ChO groups was in line with the control.

### Interpretation

Various attempts have been made to improve the ALBC efficacy. In our study, the best bone cement formulation was bone cement mixed with ChO (10%), which had the highest drug

**Funding:** S.T. received funding from the Chulalongkorn University Technology Center and Thailand Science Research and Innovation Fund (TSRI) (CU_FRB640001_01_2). A.W. received funding from the Specific League Funds from Mahidol University. The funders had no role in the study design, data collection and analysis, decision to publish, or preparation of the manuscript

**Competing interests:** The authors have declared that no competing interests exist.

release profiles, was biocompatible, and contained antibacterial properties with acceptable mechanical properties. This phenomenon could result from the superior water solubility of the ChO. When ChO leaves the bone cement specimens, it generates pores that could act as a path that exposes the bone cement matrix to the surrounding medium, increasing antibiotic elution. From all above, ChO is a promising substance that could be added to ALBC in order to increase the drug elution rate. However, more *in vitro* and *in vivo* experiments are needed before being used in the clinic.

## Introduction

Polymethylmethacrylate (PMMA) bone cement played an imminent role in treating orthopaedic infection. Antibiotic-loaded bone cement(s) (ALBC) or PMMA bone cement mix with antibiotic(s) has been used as a reservoir for local antibiotics. In the current practice, the use of ALBC is an established method in managing orthopaedic infections, such as prosthetic joint infections (PJIs) and chronic osteomyelitis [1]. ALBC can be used in the form of ALBC beads or spacer impregnated into the infected area [2].

ALBC types can generally be divided into manual mixing and commercially premixed. Although the manual mixing method provides high flexibility regarding the antibiotic numbers, types, and concentration, the manual mixing ALBC could give an unpredictable drug release profile, improper mechanical properties, and unsafe for patients [3]. The commercially premixed ALBC has several advantages, including known drug elution profile, reduced surgical time, and more homogenously mixed between antibiotics and PMMA [4]. To get superior control of pathogens, some premixed ALBC(s) are composed of two or more antibiotics. The dual ALBC has stronger antimicrobial action due to the synergistic drug release effect resulting from increasing bone cement porosity from dissolved antibiotics. These porosities increase body fluid-bone cement contact surface area and serve as a path for the body fluid to reach deeper into the bone cement bulk, increasing the elution of one or both antibiotics. This phenomenon is called "passive opportunism" [5, 6]. Although all these attempts to improve the antibiotic release from bone cement, there are reports of antibiotic resistance [7].

In recent years, several new materials have been introduced to bone cement to improve ALBC antimicrobial efficacy. Combining inorganic antimicrobial agents such as silver nanoparticles can enhance antibacterial effectiveness. Adding porogen such as glucose [8], lactose [9], and xylitol [10] to increase the porosity of the bone cement bulk is also an effective way to improve the amount of drug release from ALBC. Among these new substances, chitosan (Ch) and its derivative (chitosan oligosaccharides, ChO) have great potential to be used in bone cement.

Ch is a biodegradable and biocompatible material that can be easily degraded in the body fluid, increasing ALBC porosity and antibiotic(s) elution [11]. In addition, there is ChO, an oligomer of β-(1→4)-linked D-glucosamine, that is water-soluble and can be an even better and faster dissolver than regular Ch. Interestingly, both Ch and ChO had evidence supporting their antibacterial and antifungal effect, which would be an excellent addition to ALBC. However, adding a high concentration of these fillers into the ALBC can considerably affect its mechanical properties. The current information about the pharmaceutical and mechanical effects of applying chitosan/oligo-chitosan into the ALBC is still lacking.

This study aimed to study the effect of adding Ch and ChO into the commercially premixed antibiotic bone cement (Copal® G+V). We compare antibiotic release profiles, bacterial control activity, and mechanical properties among different ALBC formulations. The hypothesis

was that increasing the amount of Ch and ChO in the cement mixture would increase the antibiotics released and bacterial control. We believe that ALBC mixed with ChO may create a more significant effect due to its superior dissolving property.

## Materials and methods

### Materials

The bone cement samples used in this project were made from Copal® G+V (Heraeus Medical GmbH, Germany). Copal® G+V is a radiopaque revision bone cement made from PMMA. This bone cement contains 0.5 g gentamicin sulfate and 2.0 g of vancomycin hydrochloride (vancomycin HCl) in 43.0 g of the bone cement powder component.

Chitosan (Lot no. 190221/1-1) and chitosan oligosaccharide (Lot no. 190221/2-1) were purchased from the Marine Bio-Resources Co., Ltd., Thailand. Ch was obtained from shrimp (*Litopenaeus vannamei*) with a percent degree of deacetylation equal to 97.46 and average molecular weight of approximately 25 kDa. ChO was also obtained from shrimp (*Litopenaeus vannamei*) with a percent degree of deacetylation equal to 87.79 and average molecular weight $\leq$ 5 kDa. Ch and ChO were tested for pathogens and passed the standard with a total plate count $\leq$ 1,000 colony forming units (CFU)/g, and yeasts and molds $\leq$ 100 CFU/g. *Escherichia coli* contamination was less than 3 most probable number (MPN)/g with no detection of *Staphylococcus aureus*, *Clostridium* spp., *Salmonella* spp., and Coliform.

### Preparation of bone cement samples

The bone cement samples were cast in silicone molds made of poly(dimethylsiloxane) or PDMS. The mold has disc features (19.5 mm in diameter with 3.5 mm in height). The control group was made purely with Copal® G+V by hand-mixed the PMMA powder with the liquid monomer according to the manufacturer's recommendation and hand-pressed into the silicone mold.

To prepare the Ch and the ChO mixed bone cement samples, different amounts of Ch and ChO were added to the PMMA powder according to Table 1. Three concentrations (1%, 5%, and 10%w/w) of Ch and ChO were incorporated into the polymer powder using the geometric dilution technique to ensure homogeneous mixing with the polymer powder. Then, the liquid monomer was added to form a ductile dough. The dough was pressed into the prepared mold before being cured. The bone cements were allowed to cure for 24 h before further testing. Size (diameter and thickness) was measured using a digital vernier caliper (RS Components Co., Ltd., Thailand).

**Table 1. Bone cement formulations tested in this project by mixing the poly(methyl methacrylate/methacrylate) powder with different amounts of chitosan or chitosan oligosaccharide.**

| Formulation number | Formulation name | Chitosan content (%w/w) | Chitosan oligosaccharides content (%w/w) | Chitosan content (g) | Chitosan oligosaccharide content (g) | Copal® G+V powder | Total weight (g) |
|---|---|---|---|---|---|---|---|
| 1 | Control | - | - | - | - | 43.00 | 43.00 |
| 2 | Ch 1% | 1 | - | 0.43 | - | 42.57 | 43.00 |
| 3 | Ch 5% | 5 | - | 2.15 | - | 40.85 | 43.00 |
| 4 | Ch 10% | 10 | - | 4.30 | - | 38.70 | 43.00 |
| 5 | ChO 1% | - | 1 | - | 0.43 | 42.57 | 43.00 |
| 6 | ChO 5% | - | 5 | - | 2.15 | 40.85 | 43.00 |
| 7 | ChO 10% | - | 10 | - | 4.30 | 38.70 | 43.00 |

Ch: Chitosan; ChO: Chitosan oligosaccharides, %w/w: Percentage weight by weight.

## Drug elution assay

A drug elution assay was conducted in prepared bone cement samples. All seven formulations (Table 1) of bone cement samples were placed in centrifuge tubes containing 30 mL of phosphate-buffered saline (PBS ultra-pure grade, 0.1 μm sterile filtered, Apsalagen, Thailand). Samples were stored in an incubator shaker at 37°C, 100 rpm. At designated times (1, 3, 6, 24 h, 2 3 5, and 7 days), 1,000 μL of samples were removed and replaced with pre-warmed PBS. All samples (PBS-containing antibiotics) were stored at -20°C for further analysis. The supernatant obtained at each time point was analyzed for antibiotic concentration and antimicrobial activity. At the end of the experiment (7-day period), bone cements were removed and gently rinsed with sterile water for irrigation (SWI, General Hospital Products Public Co., Ltd., Bangkok, Thailand) before oven-dry (Hot air oven, Memmert, Germany) at 45°C overnight for further physical and mechanical properties evaluation.

Liquid samples obtained from the supernatant were quantified for vancomycin and gentamicin using the spectrophotometric and reversed-phase high-performance liquid chromatographic (RP-HPLC) methods, respectively. Vancomycin was quantified based on a coupling reaction between the drug and diazotized procaine. A quantification method by Hadi with modification was applied [12]. Procaine mixture is composed of 1.0 mM procaine hydrochloride (Sigma Life Science, USA), 1.0 mM sodium nitrite (NaNO$_2$, Carlo ERBA Reagents, USA), and 1.0 M hydrochloric acid (HCl, Chem-Lab, Belgium) was prepared. The reaction was done on a 96-well plate. Each well was composed of vancomycin HCl at different concentrations (100 μL), procaine mixtures (30 μL), and 0.125 M ammonium hydroxide (NH$_4$OH, Carlo ERBA Reagents, USA) (120 μL). The intense yellow color azo dye was measured at 447 nm using a microplate reader (ClarioStar Multimode Microplate Reader, BMG Labtech, Germany). Vancomycin hydrochloride (Lot. A4230021, kindly gifted from Siam Bheasach Co., Ltd., Bangkok, Thailand) was used as a standard. Blank was prepared the exact same way as samples but used 100 μL of PBS. Vancomycin concentrations in samples were compared against the vancomycin standard curve. Cumulative release profiles of vancomycin from bone cement samples (y-axis) were plotted against time (x-axis).

The quantitative determination of gentamicin was performed HPLC on Shimadzu HPLC system equipped with DGU-20A5R degassing unit, Prominence LC-20AD pumping system, SIL-10AD VP autoinjector, and SPD-10A VP UV/VIS detector (Shimadzu Scientific Instruments, Kyoto, Japan). The integration and system parameters were controlled by LC solution software (Shimadzu Scientific Instruments, Kyoto, Japan). The HPLC condition was modified from Blanchaert et al. [13]. An ACE column (C18; 150 mm × 4.6 mm i.d., particle size 5 μm; Advanced Chromatography Ltd., UK) was used as a stationary phase. Elution was performed using isocratic mode with 0.5 mL/min flow rate at 40°C. The mobile phase consisted of methanol (HPLC grade, Honeywell Burdick & Jackson, Ulson, Korea), disodium tetraborate decahydrate buffer (0.1 M; pH 9.0), and water containing 1 g/L sodium octane sulfonate (Sigma-Aldrich, USA) at the volume ratio of 19:10:61. The disodium tetraborate decahydrate buffer was prepared using 1-octanesulfonic acid sodium salt (Sigma-Aldrich, Missouri, United States), sodium tetraborate (Ajax Chemicals, New South Wales, Australia), and phosphoric acid (J. T. Baker Chemical, Pennsylvania, USA). The total run time was 40 min, and the sample injection volume was 40 μL. UV detection was performed by monitoring the absorbance signal at 205 nm. The HPLC method was validated for its linearity, precision, accuracy, the limit of detection (LOD), and limit of quantitation (LOQ) as per the International Conference on Harmonization (ICH) Q2 guidelines [14]. Gentamicin injection (80 mg/2 mL) (Grammicin injection, lot 1072062, Siam Bheasach Co., Ltd., Bangkok, Thailand) was kindly gifted from the Pharmacy Department, Faculty of Medicine Siriraj Hospital, Mahidol University (Bangkok,

Thailand). The gentamicin calibration curve was in the concentration ranging from 0.78 to 100 μg/mL. The calibration curve linear equation was y = 13930x-8615, and the linear correlation coefficient was 0.9999. The intra-day and inter-day precision of gentamicin was evaluated using three concentrations (3.125, 12.5, and 50 μg/mL), and the coefficient of variation was lower than 2%. The method accuracy was determined using a recovery study conducted at three concentrations (3.125, 12.5, and 50 μg/mL), and the average recovery was 99.43 ± 0.55%. The LOQ and LOD were 0.78 and 0.20 μg/mL, respectively.

## Bone cement physical characterization

Bone cement weight was quantified using a balance (Analytical Balances CP225D, Sartorius, Germany). Bone cements were imaged using a stereomicroscope (model sZ61, Olympus, Japan) using a 6.7x magnification. Surface morphology at the micron level was identified using scanning electron microscopy (SEM). Briefly, the bone cement samples were mounted on SEM stubs and then coated with gold by a sputter coater (Blazers SCD 040, Bal-Tec AG, Blazers, Liechtenstein) for three min. Images were captured using JSM-IT300 InTouchScope™ SEM (JEOL, Tokyo, Japan) at 10.0 kV accelerating voltage. All physical characterizations were conducted with bone cement before and after the drug elution assay.

## Determination of antibacterial activity

The antibacterial activity of vancomycin and gentamicin eluted from bone cement specimens was estimated using an agar disc diffusion method according to M100 Performance standards for antimicrobial susceptibility test, 30th edition, The Clinical and Laboratory Standards Institute (CLSI). Liquid samples were supernatant obtained from the drug elution assay at the end of the experiment (day 7). Twenty microlitres of samples were added to the filtered papers. The tested papers were placed on Mueller Hinton Agar (Difco™, Maryland, USA) inoculated with *S. aureus* ATCC® 25923 or methicillin-resistant *S. aureus* (MRSA DMST 20654) in a density of $1.5 \times 10^8$ CFU/mL (adjusted with 0.5 McFarland turbidity standard). The samples were incubated at 35±2˚C for 16–20 h. At the end of the experiment, the zone of inhibition (ZOI) was measured and reported in mm units. PBS was used as a negative control. Vancomycin (30 μg per disc) was used as a positive control. Each plate contains three samples, one negative and one positive control.

## *In vitro* cytotoxicity assay

The human osteosarcoma cell line (Saos-2, HTB-85™) was kindly gifted by Dr. Pakpoom Kheolamai, Division of Cell Biology, Faculty of Medicine, Thammasat University, Thailand. Cells were maintained in Dulbecco's Modified Eagle's Medium (DMEM, Gibco®, Life Technologies, USA). The media was supplemented with 10% fetal bovine serum (FBS, triple 0.1 μm sterile filtered, Hyclone™, GE Healthcare Bio-Sciences, Austria) with 1% penicillin/streptomycin (Gibco®). Cells were incubated at 37˚C and 5% $CO_2$. Sub-culturing was performed between 1:2 to 1:4 ratio before cells reached 80% confluency.

An evaluation for *in vitro* cytotoxicity was conducted according to ISO 10993–5, Biological evaluation of medical devices, Part 5 with some modifications. The treatments used in this experimental set were liquid extracts of bone cement using PBS as an extraction vehicle. The extraction was conducted by incubating bone cement in PBS for 24 h at 37˚C, one of the conditions used to measure the hazard potential for risk estimation of the medical device/material.

Cell viability is based on the measurement of cells' metabolic activity that reduced the yellow MTT (3-(4,5-dimethylthiazol-2-yl)-2,5-diphenyltetrazoliumbromide) to a violet formazan. Saos-2 cells were plated 24 h before the treatment in 96-well plates at a concentration of $1 \times 10^4$

cells per well. Treatments (50 μL) were added into each well along with fresh medium (50 μL) for 24 h. At the end of the incubation period, the microscopic evaluation of morphological alterations was conducted, and the treatments were removed. The MTT solution (PanReac AppliChem ITW Reagents, Spain) in the serum-free medium at the concentration of 0.5 mg/mL was then added to each well. After 2 h of incubation at 37°C, 5% $CO_2$, the media were removed. Dimethyl sulfoxide (DMSO, Sigma-Aldrich, USA) was added to dissolve the formazan crystals. The absorbance was recorded at 570 nm using a microplate reader (ClarioStar Multimode Microplate Reader, BMG Labtech, Germany). Cell viability was expressed as a percentage of the absorbance value of cells that were treated with PBS. All absorbance values were corrected with a blank solution that contains 100 μL of DMSO.

Sodium lauryl sulfate (SLS, S. Tong Chemicals Co., Ltd., Thailand) was introduced as a positive control. Cells were treated with SLS at 4 concentrations ranging from 25–100 μg/mL for 24 h.

## Mechanical properties testing

**Measuring surface roughness.** The surface roughness of bone cements in all groups was assessed using Alicona® Infinite Focus SL (Alicona, Austria) 3D measurement system. Firstly, the images were taken at the 10x optic with a contrast of 2.4 and a vertical resolution of 200 nm. The polynomial is then removed from the form before measuring the profile's form using all-points mode. Surface roughness was characterized by the average roughness (Ra, μm) obtained from a 4-mm profile length (n = 5).

**Measuring microhardness.** All bone cements were polished before microhardness assessment for more accurate results. The polishing was conducted in 4 steps using different sandpaper abrasive grits from P400, P800, P1200, and velvet sandpapers for 40 s, 100 s, 100 s, and 150 s, respectively, at 300 rpm. For P400, P800, and P1200 grit, the water was used as a grinding media. For velvet sandpapers, the water containing 0.05 μm alumina powder was used as a grinding media.

The microhardness test that was used with bone cement samples was the Vickers hardness test using a Vickers indenter pressed into a polished bone cement surface to a specified force. The Vickers hardness test was conducted using a hardness measurement machine (FM-810, Future-Tech, Japan) with a test load of 100 gf, dwell time of 10 s, and measured the lengths of the diagonals to determine the impression size under a 20x lens. The Vickers hardness (HV, kgf/mm$^2$) was calculated according to the following equation:

$$HV = \frac{1.8544\,F}{d^2},$$

Where F is the force provided by the machine (kgf), and d is an average length obtained from an impression on the x-axis and y-axis (mm).

## Statistical analysis

Data are expressed as mean ± standard error of the mean (SEM). Statistical significance was determined using One-way ANOVA with Dunnett's multiple comparisons test (to compare the mean of the control). Two-way ANOVA with Dunnett's multiple comparisons tests was performed to compare the difference with the control group. Two-way ANOVA with Sidak multiple comparisons tests was performed to compare the difference within the same group, before and after the drug elution test. All statistical tests were performed using GraphPad Prism version 7.00 (GraphPad Software, CA, USA, www.graphpad.com). A $p$-value less than 0.05 was considered significant.

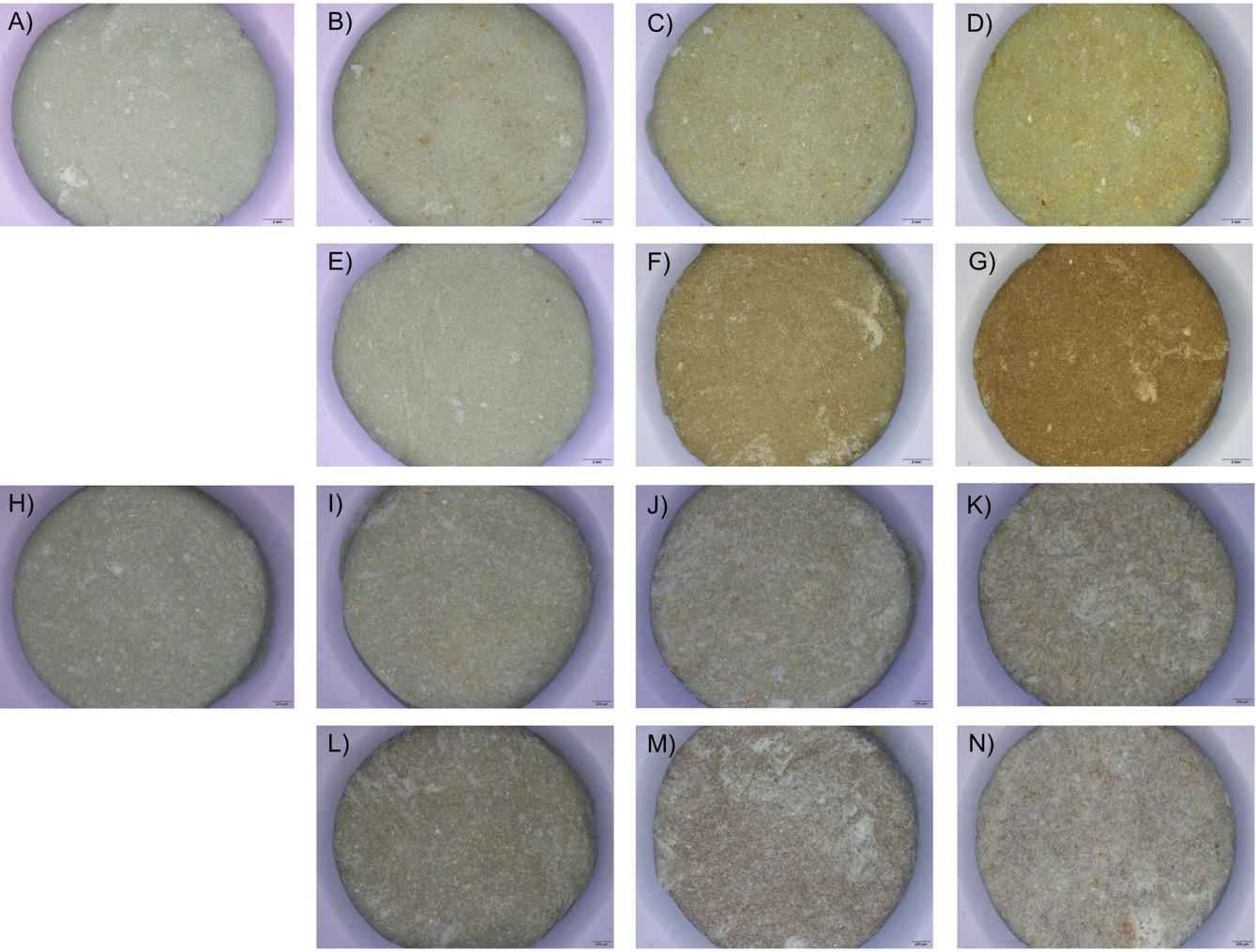

**Fig 1.** Freshly prepare (A-G) and drug eluted (H-N) bone cements images obtained from a stereomicroscope with 6.7x magnification. Bone cement made solely from Copal® G+V (Control, A and H); Copal® G+V mixed with 1% (Ch 1%: B and I), 5% (Ch 5%: C and J), and 10% (Ch 10%: D and K) w/w chitosan; Copal® G+V mixed with 1% (ChO 1%: E and L), 5% (ChO 5%: F and M), and 10% (ChO 10%: G and N) w/w chitosan oligosaccharides. Scale bar represents 2 mm.

## Results

### Bone cement preparation and characteristic

Bone cement specimens were successfully fabricated using PDMS molds. The diameter (S1A Fig) and thickness (S1B Fig) of the bone cement samples were approximately 18.3 ± 0.1 mm and 5.3 ± 0.1 mm, respectively. The bone cement weight is approximate 1.44 ± 0.01 g per piece. There are no significant differences between diameter, thickness, and weight between the control, Ch, and ChO groups.

Ch powder has a light-yellow color. ChO has a dark yellow color. Thus, incorporating Ch and ChO into Copal® G+V (light green color) resulted in color change. According to Fig 1, bone cement samples turned yellow when the Copal® G+V mixed with Ch (Figs 1B and 1C and 2D). The color profile was distinct, especially when a high percentage of Ch was used. Since ChO has a darker yellow color when compared to Ch, Copal® G+V incorporated ChO (Fig 1E–1G) had darker yellow than Copal® G+V mixed with Ch. Among all samples, ChO

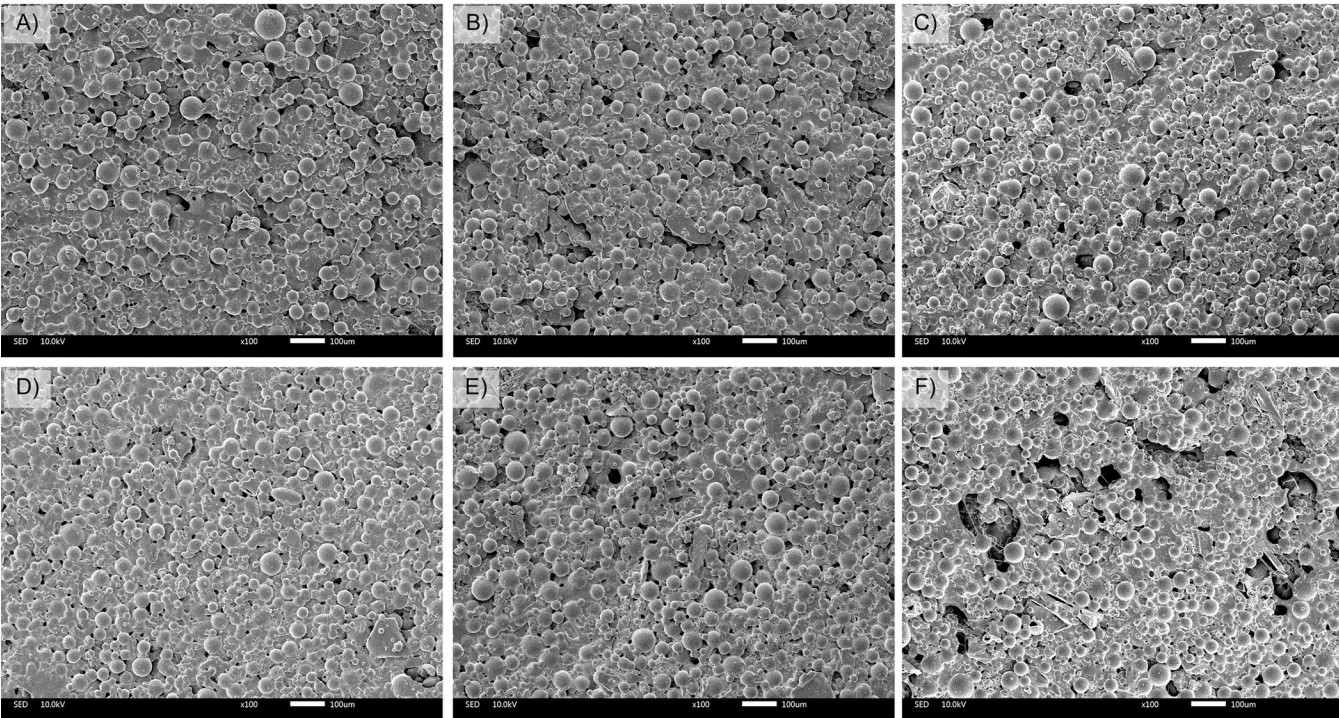

**Fig 2.** SEM micrographs represent bone cement samples obtained after freshly prepared (A-C) and after submerged in PBS for 7 days (D-F). Bone cement made solely from Copal® G+V (Control: A, D); Copal® G+V mixed with 10%w/w chitosan (Ch 10%: B, E); Copal® G+V mixed with 10%w/w chitosan oligosaccharides (ChO 10%: C, F). Scale bar represents 100 μm.

10% or Copal® G+V mixed with 10%w/w chitosan oligosaccharides had the darkest color (Fig 1G).

Physical characterization was conducted after the drug elution assay. From the stereomicroscope images, all bone cement samples showed light green color, which is the original color of Copal® G+V. There is no yellow color which can infer that most of the Ch and ChO on the cement surface were eluted into PBS (Fig 1H–1N). To observe the changes in surface morphology, bone cement mixtures with the highest percentage of Ch and ChO were imaged using SEM. A few pores were detected in Ch 10% (Fig 2E) and ChO 10% (Fig 2F) after submerging in PBS for 7 days. When Ch and ChO groups were compared, ChO 10% exhibited a higher porosity than Ch 10%. Although there were changes in bone cement appearance (from stereomicroscope) and surface morphology (from SEM), there was no significant change in weight after the submerge (Fig 3).

## Drug release profile

The main purpose of mixing Ch and ChO into bone cement is to increase the antibiotic concentration released into the surrounding. In this study, both antibiotics in bone cement, which are vancomycin (in the form of vancomycin hydrochloride) (Fig 4) and gentamicin (in the form of gentamicin sulfate) (Fig 5), were quantified and expressed as the cumulative drug release (μg) versus time (h).

Bone cement samples made from Copal® G+V alone or combined with Ch or ChO can release vancomycin and gentamicin into the PBS. The release of vancomycin was rapid in the first 48 h and continued at a much lower rate (reaching the plateau phase) afterward. These release profile patterns were similar in all groups tested. There is no significant difference

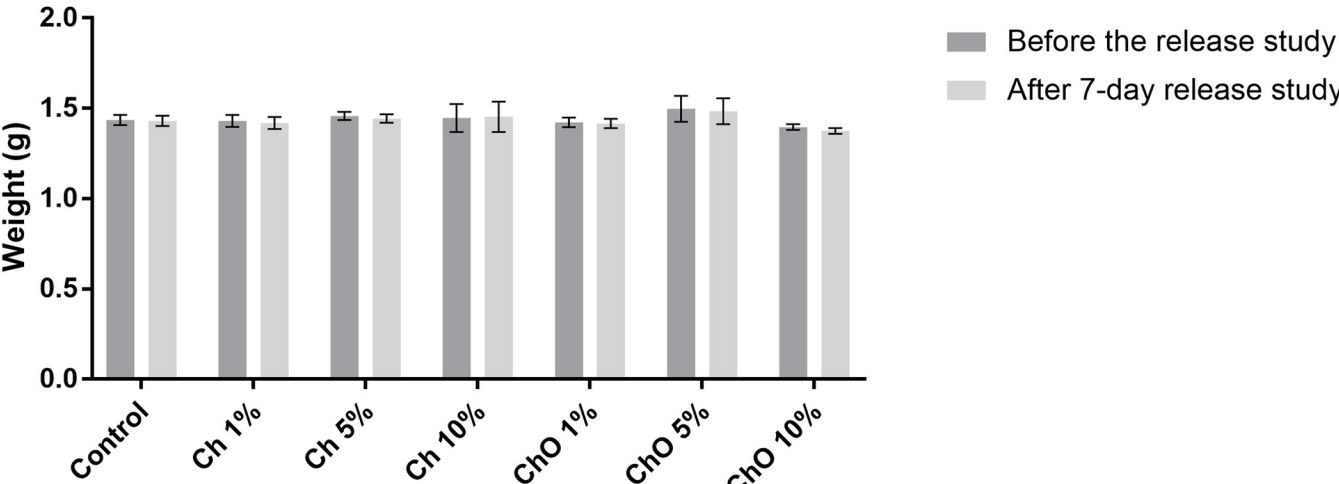

**Fig 3. Weight of bone cements obtained after casting bone cement mixtures before and after drug elusion.** Control represents bone cement specimens made solely from Copal® G+V. Ch 1%, Ch 5%, and Ch 10% are specimens made of Copal® G+V mixed with 1%, 5% and 10% w/w chitosan, respectively. ChO 1%, ChO 5%, and ChO 10% are specimens made of Copal® G+V mixed with 1%, 5% and 10% w/w chitosan oligosaccharides, respectively. Data are expressed as mean ± SEM (n = 6). Two-way ANOVA with Dunnett's multiple comparisons test was performed.

between the vancomycin release profiles from the control group and cement mixed with chitosan (Ch 1%, Ch 5%, and Ch 10%) (Fig 4A) at each timepoint. However, after 48 h, the cumulative vancomycin release profiles from ChO 5% and ChO 10% were higher than the control group with a significant difference ($p < 0.001$) (Fig 4B).

The gentamicin release profile was time-dependent and increased almost in a linear manner (Fig 5). These linear patterns were detected in all groups tested. According to Fig 5A, there is no significant difference between the gentamicin release profiles from the control group and the Ch group (Ch 1%, Ch 5%, and Ch 10%). The exception was found in the gentamicin amount released from Ch 5% and Ch 10% at 7 days of the drug elution assay ($p < 0.05$). For cement mixed with ChO (Fig 5B), the cumulative gentamicin release profiles of ChO (ChO 1%, ChO 5%, and ChO 10%) were significantly higher than the control group at 72 h until 7 days ($p < 0.001$) of the drug elution assay.

Considering the amount of drug released at each time point, the bone cement samples in all formulations tested released gentamicin in a lower amount than vancomycin. At 7 days, Ch and ChO groups released vancomycin approximately 3,000–3,500 μg and 3,200–7,000 μg per piece, respectively. At 7 days, the amount of gentamicin released from the Ch and ChO groups was approximately 1,600–2,000 μg and 2,500–2,800 μg per piece, respectively. The lower drug release profile from gentamicin is as expected since Copal® G+V contains vancomycin 4-fold higher than gentamicin (by weight).

## Antibacterial activity

*S. aureus* [15, 16] and MRSA [17–19] are the most common pathogen in orthopaedic surgical site infections. The antimicrobial activity of bone cement specimens was tested against *S. aureus* and MRSA using supernatant obtained from the drug elution assay at the end of the experiment (day 7). ZOI of *S. aureus* (S2 Fig) and MRSA (S3 Fig) were measured. Comparing two types of bacteria, the supernatant obtained in all samples incubated with *S. aureus* (25–30 mm, Fig 6A) gave a larger ZOI than that incubated with MRSA (10–15 mm, Fig 6B). *S. aureus* was shown to be more susceptible to the drug mixture eluted from Copal® G+V than MRSA, which is expected [20, 21].

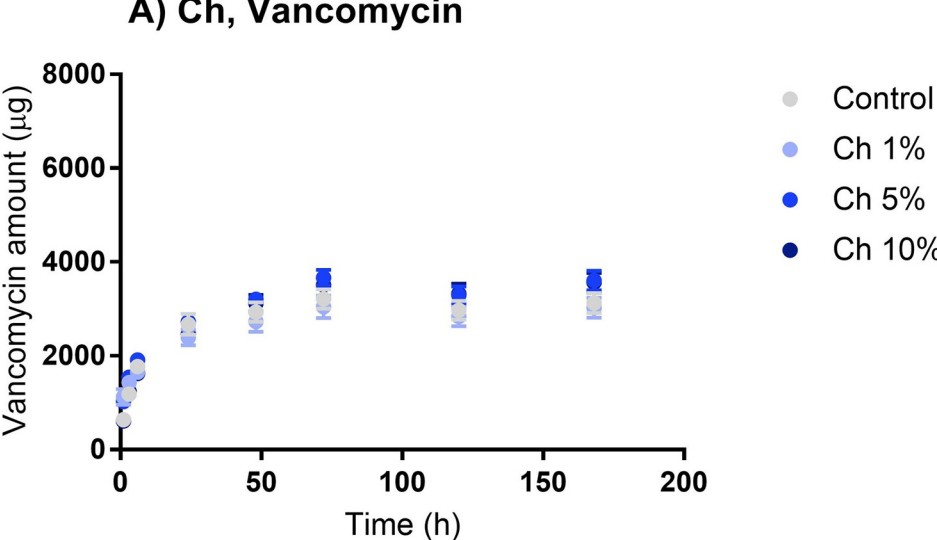

**Fig 4.** Cumulative release of vancomycin from bone cement prepared with chitosan (A) and chitosan oligosaccharides (B). Control represents bone cement specimens made solely from Copal® G+V. Ch 1%, Ch 5%, and Ch 10% are specimens made of Copal® G+V mixed with 1%, 5% and 10% w/w chitosan, respectively. ChO 1%, ChO 5%, and ChO 10% are specimens made of Copal® G+V mixed with 1%, 5% and 10% w/w chitosan oligosaccharides, respectively. Data are expressed as mean ± SEM (n = 6). Two-way ANOVA with Dunnett's multiple comparisons test was performed. ***$p < 0.001$.

The results showed that all samples could inhibit these bacteria. Incorporating Ch or ChO into the bone cement matrix did not affect the antibacterial activity of Copal® G+V. In contrast, they can increase the antibacterial activity of the bone cement samples. ZOI in each group is consistent with the amount of vancomycin (Fig 4) and gentamicin (Fig 5) eluted. The supernatant obtained from bone cement samples with the highest amount of drug eluted (ChO 10%) gave the largest ZOI.

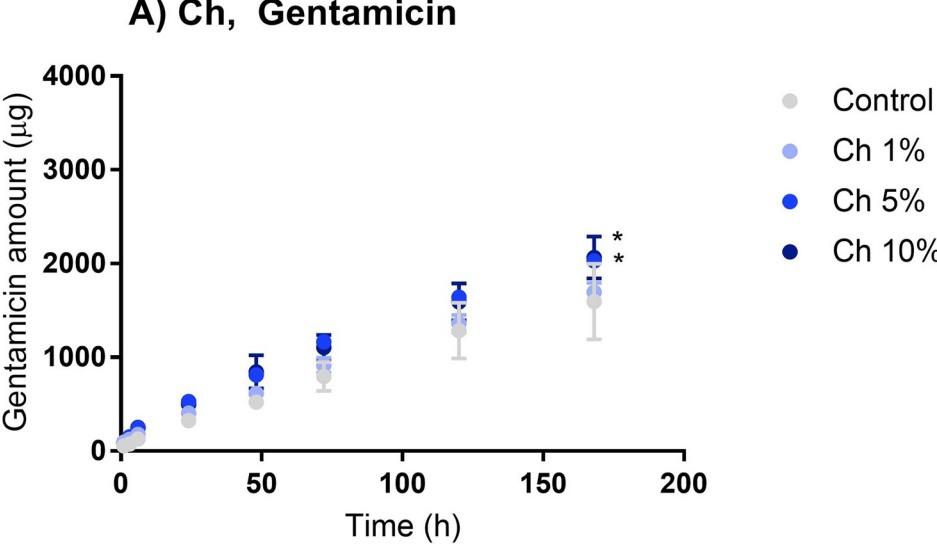

**Fig 5.** Cumulative release of gentamicin from bone cement prepared with chitosan (A) and chitosan oligosaccharides (B). Control represents bone cement specimens made solely from Copal® G+V. Ch 1%, Ch 5%, and Ch 10% are specimens made of Copal® G+V mixed with 1%, 5% and 10% w/w chitosan, respectively. ChO 1%, ChO 5%, and ChO 10% are specimens made of Copal® G+V mixed with 1%, 5% and 10% w/w chitosan oligosaccharides, respectively. Data are expressed as mean ± SEM (n = 3). Two-way ANOVA with Dunnett's multiple comparisons test was performed. $^*p<0.05$, $^{**}p<0.01$, $^{***}p<0.001$.

### *In vitro* cell cytotoxicity

An MTT assay was used to assess the cytotoxicity of the bone cement and its degradation products. Saos-2 cells were exposed to media obtained from bone cement extractions for 24 h. For the qualitative morphological grading of the extracts' cytotoxicity, all samples received a score of 0. There is no reactivity. There was no cell lysis or decrease in cell proliferation, and no intracytoplasmic granules were detected. All samples tested showed a % cell viability higher

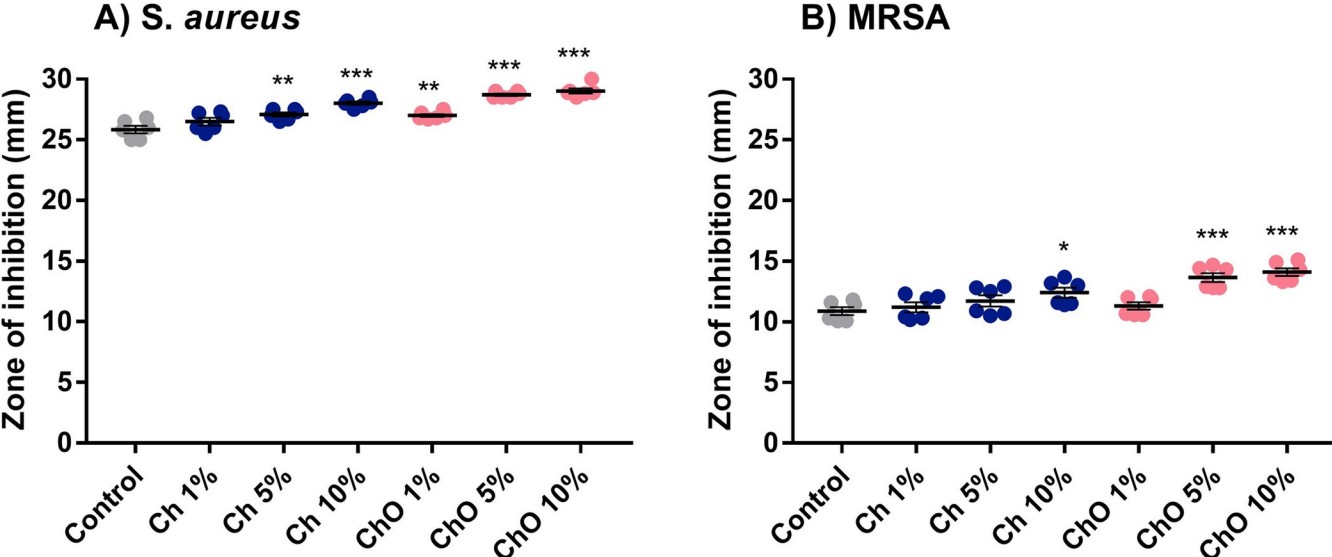

**Fig 6.** Zone of inhibition (ZOI) of *S. aureus* (A) and MRSA (B) of the supernatant obtained from different bone cement specimens that incubated in PBS for 7 days. Control represents bone cement specimens made solely from Copal® G+V. Ch 1%, Ch 5%, and Ch 10% are specimens made of Copal® G+V mixed with 1%, 5% and 10% w/w chitosan, respectively. ChO 1%, ChO 5%, and ChO 10% are specimens made of Copal® G+V mixed with 1%, 5% and 10% w/w chitosan oligosaccharides, respectively. Data are expressed as mean ± SEM (n = 6). One-way ANOVA with Dunnett's multiple comparisons test was performed.

than 70, which is considered no cytotoxic effect according to ISO 10993–5 (Fig 7). In this experiment, SLS was used as a positive control in concentrations ranging from 25 to 100 µg/mL (S4 Fig). SLS-treated cells exhibited concentration-dependent toxicity.

## Mechanical properties

Roughness and microhardness of bone cements were tested (Fig 8). In the freshly prepared group, the average roughness was between 5 to 6 µm. The surface of bone cements containing Ch was rough, especially when a large quantity of Ch was incorporated (Fig 8A). Ch 10% had roughness equal to 9.66 ± 2.23 µm ($p < 0.01$). However, there is no difference in surface roughness between the control and ChO groups. After a 7-day release study, the surface roughness was reduced in all groups tested. This reduction is no significant difference when compared before and after the release study ($p > 0.05$).

For the microhardness measurement of freshly prepared bone cements, the control group had a microhardness value of approximately 15.49 ± 0.44 kgf/mm$^2$, while others had higher microhardness values (Fig 8B). However, after a degradation period of 7 days, there was a significant reduction ($p < 0.001$) in microhardness in all groups tested.

## Discussion

ALBC is a common strategy used to treat orthopaedic infections as complementary to the systemic antibiotic(s) administration. Even though ALBC has been in use for a long time, its efficacy is not ideal. The drug release is often uncontrollable and would give rise to drug-resistant microbial strains. Several efforts were made to enhance the ALBC efficacy. In our study, the influence of Ch and ChO when incorporated into the bone cement was investigated. Mixing hydrophilic, biocompatible, and biodegradable polymers into bone cement would increase the antibiotics released and better bacterial control. This study chose the commercially available, Copal®G+V containing dual antibiotics (vancomycin and gentamicin) as the bone cement

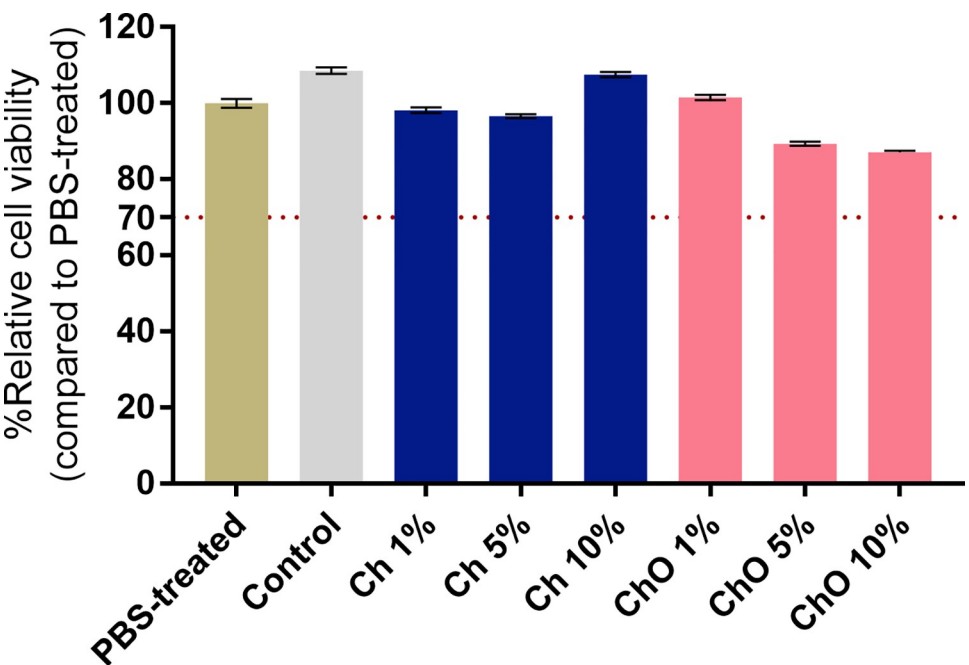

**Fig 7. Relative cell viability (%) of Saos-2 cells after treatment with extracts from various bone cement samples for 24 h.** Control represents bone cement specimens made solely from Copal® G+V. Ch 1%, Ch 5%, and Ch 10% are specimens made of Copal® G+V mixed with 1%, 5% and 10% w/w chitosan, respectively. ChO 1%, ChO 5%, and ChO 10% are specimens made of Copal® G+V mixed with 1%, 5% and 10% w/w chitosan oligosaccharides, respectively. PBS-treated group was considered 100% cell viability. Data are expressed as mean ± SEM (n = 36–41).

matrix. Ch and ChO were mixed into the bone cement powder before forming bone cement dough with a liquid monomer at the weight ratio of 1%, 5%, and 10%. Since chitosan and chitosan oligosaccharides were yellow, the bone cement color changed toward yellow after mixing (Fig 1). These color changes were directly correlated with the amount of Ch and ChO. The drug elution was performed to determine the drug release profiles from each specimen. The yellow color in Ch and ChO groups faded after being submerged in PBS for 7 days. The change in appearance and the noticeable pores on Ch and ChO surfaces (from SEM images) infer that Ch and ChO dissolved out of the bone cement samples.

Bone cement samples made from Copal® G+V alone or combined with Ch or ChO can release vancomycin and gentamicin into the PBS. Mixing ChO into the bone cements can increase the amount of drug released more than Ch. ChO 10% gave the highest amount of antibiotics released. The vancomycin release profile pattern from Copal® G+V bone cement specimens was similar to other published works that consisted of a rapid release in the first 48 h followed by a plateau phase [22–24]. For example, Lee et al. [22] studied vancomycin release from various bone cement specimens available in the market (Surgical Simplex P, Osteobond, Palacos® R, and Dupey-CMW). Although different types of bone cement release different amounts of vancomycin, the vancomycin concentrations in all samples studied reached a plateau phase after 72 h [22].

In contrast, the gentamicin release profile patterns varied among researchers worldwide. Some researchers found that the gentamicin release profile reached a plateau after 72 h of the drug elution assays [24, 25]. Some researchers reported that the gentamicin release profile increased in the buffered solution beyond 72 h [26]. Ensing et al. reported the gentamicin release profile in PBS, which was increased almost linearly in the medium after a long time

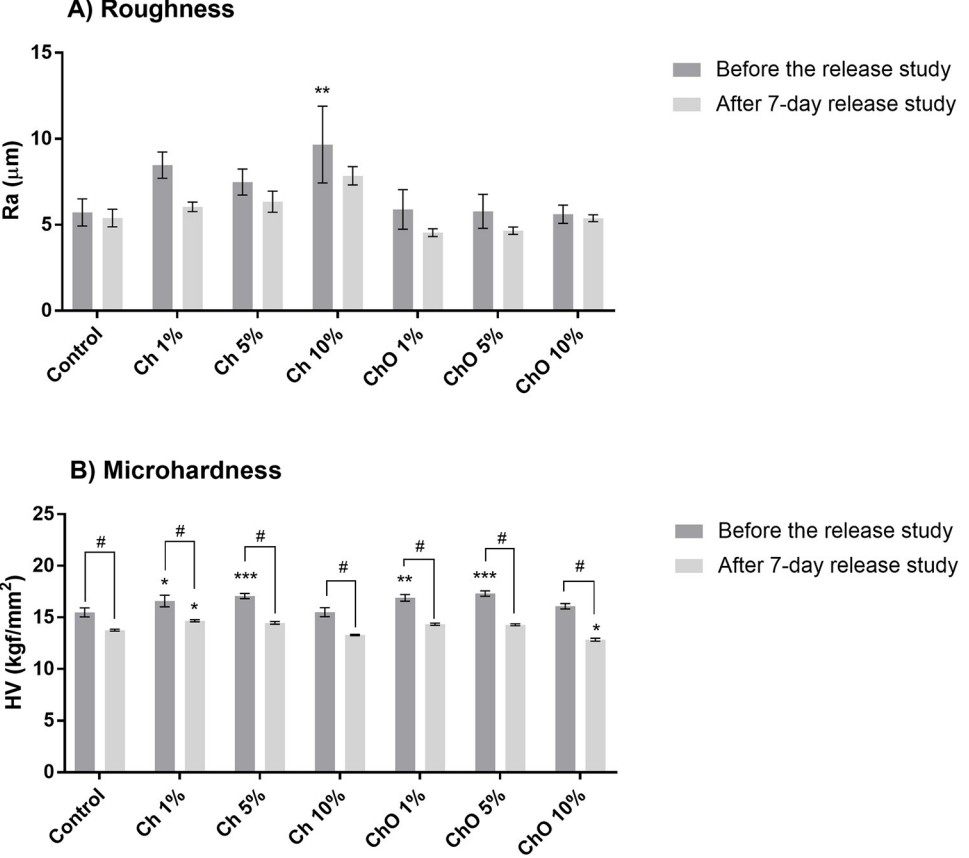

**Fig 8.** Roughness (μm) (A) and microhardness (HV) (B) of various bone cement samples before and after the drug release study. Control represents bone cement specimens made solely from Copal® G+V. Ch 1%, Ch 5%, and Ch 10% are specimens made of Copal® G+V mixed with 1%, 5% and 10% w/w chitosan, respectively. ChO 1%, ChO 5%, and ChO 10% are specimens made of Copal® G+V mixed with 1%, 5% and 10% w/w chitosan oligosaccharides, respectively. Data are expressed as mean ± SEM (n = 5–6). The statistical different between group were analyzed using Two-way ANOVA with Dunnett's multiple comparisons test comparing to the control group. $*p<0.05$, $**p<0.01$, $***p<0.001$. The statistical different in the same bone cement samples before and after the release study was analyzed using Two-way ANOVA with Sidak multiple comparison test. $^{\#}p<0.001$.

[26]. The gentamicin release profiles from Ensing et al. were similar to the release profiles obtained in the current study. This controversy could come from the fact that each study used bone cements from different companies. Other variables such as size, shape, the surface of bone cement, and the possible interaction between the drug molecule and PMMA or other ingredients in the formulation could affect the drug release profile.

This study aims to investigate the amount of drug released from bone cement specimens when combined with Ch or ChO. Mixing Ch into bone cement powder before adding liquid monomer resulted in a slight increase in the vancomycin and gentamicin release (Figs 4A and 5A). However, there was a significant increase in the drug release profiles when bone cement powder was mixed with ChO (Figs 4B and 5B). Among all groups, ChO 10% showed the highest drug release. This difference could result from the superior water solubility of the ChO. Ch is readily soluble in dilute acid mixtures such as dilute acetic acid but is insoluble in water [27]. On the other hand, the ChO is readily soluble in water at neutral pH because of the shorter chain length with free amino groups in D-glucosamine units [28]. It is highly likely that when the ChO leaves the bone cement specimens, it generates pores. This assumption was supported

by SEM images in Fig 2, which indicated that ChO 10% had a porous surface compared to other groups. These pores could serve as a path that exposes the bone cement matrix to the surrounding medium, increasing antibiotic elution. From the results, the amount of drug released was directly correlated to the amount of Ch or ChO in the bone cement mixture. Ch 10% released the antibiotics at a higher rate than Ch 1%. Increasing the Ch or ChO concentration from 1% to 10% could increase the pores and create more interconnected pores within the bone cement matrix, thus increasing the drug elution.

Antibacterial property is the most crucial property of ALBC. Moreover, evidence suggests that Ch and ChO are antibacterial. Although the amounts of vancomycin and gentamicin were quantified, measuring antibacterial activity is necessary. The supernatant obtained from all bone cement specimens can inhibit *S. aureus* and MRSA. ZOI size was directly correlated with the antibiotics analyzed. Besides the vancomycin and gentamicin, the antimicrobial activity could come from Ch and ChO that could elute from the bone cements. This theory was supported by several publications [29–31]. According to Goy et al., Ch has broad-spectrum antibacterial and antifungal activity [29]. Ch can enhance the antibacterial activity of other materials [32]. ChO can inhibit *S. aureus* [33], *S. agalactiae* [34], and *Vibrio vulnificus* [35]. There were reports that ChO can inhibit biofilm formation [34] and could act as an antiparasitic agent [36].

Besides the antibacterial activity and biocompatibility of the Ch and ChO bone cement mixture, the study of the mechanical properties of the bone cement is crucial since adding other substances into bone cements could reduce mechanical properties [37]. Mixing Ch into bone cements increased roughness (Fig 8A). It is possible that a high roughness surface would result from the large particle size of Ch. However, this phenomenon was not detected in the ChO group since ChO powder was smaller than Ch powder. The microhardness values of the Ch and ChO groups increased significantly compared to the control group (Fig 8B). This is consistent with the work by Chander et al. that Ch can increase flexural strength, fracture toughness, and impact strength of the PMMA composite [38]. In all groups tested, the microhardness of bone cements was reduced after the drug eluted out. However, this reduction of the Ch and ChO groups was in line with the control. Adding Ch and ChO at the maximum of 10% would not compromise the microhardness of the material.

This study proved that mixing ChO into bone cement mixtures can increase the amount of vancomycin and gentamicin released into the medium. Furthermore, the antibacterial property of ChO groups was directly correlated with the amount of ChO in the bone cements. Surprisingly, this phenomenon was not observed when the bone cement was combined with Ch. From above, ChO appears to be a promising substance that could be developed and studied further as a bone cement mixture in the future.

This study has several limitations. First, ChO was used in this study to increase the drug elution. There are no reports of its clinical application as a component in an orthopaedic material. However, there are reports of ChO being used as a hydrogel in drug delivery and bone tissue engineering [39]. ChO promotes osteoclast formation [40]. As a result, it is expected that using ChO as a bone cement mixture will be safe and feasible. Second, the amount of Ch and ChO released in the medium did not quantify. Additionally, the antibacterial properties of pure Ch and ChO against *S. aureus* and MRSA have not been tested. More research is needed to demonstrate the antibacterial properties of Ch and ChO in this context. Third, only one brand of bone cement (Copal® G+V) was used in this study. Using different bone cement brands may result in a different drug release profile pattern. Forth, hand mixing was used to blend Ch or ChO with PMMA powder. Although there are reports suggesting that mechanical mixing would give a more homogeneous mixture, the geometric dilution technique was introduced to obtain the best mixture possible. In addition, this hand mixing represents a real-life scenario

in the operating room. We submit, therefore, that none of these limitations undermines the validity of our study findings.

## Conclusion

Various attempts have been made to improve the efficacy of ALBC. Adding porogen to increase the bone cement porosity can increase the drug elute from ALBC. In our study, Ch and ChO were chosen due to their potential antimicrobial activity, abundance in nature, low cost, and biocompatibility. To the best of our knowledge, this is the first study that observed the influence of Ch and ChO on ALBC using dual antibiotics. According to the findings, adding Ch and ChO to the bone cement matrix can enhance the drug release. ChO 10% was the best bone cement formulation with high drug release, effective against the common pathogen found in orthopaedic, biocompatible, and with acceptable mechanical properties. ChO is a promising substance that could be added to ALBC to increase the drug elution rate. However, more *in vitro* and *in vivo* experiments are needed before being used in the clinic.

## Supporting information

**S1 Fig.** Diameter (A) and thickness (B) of bone cement samples obtained after casting bone cement mixtures in the mold. Control represents bone cement specimens made solely from Copal® G+V. Ch 1%, Ch 5%, and Ch 10% are specimens made of Copal® G+V mixed with 1%, 5% and 10% w/w chitosan, respectively. ChO 1%, ChO 5%, and ChO 10% are specimens made of Copal® G+V mixed with 1%, 5% and 10% w/w chitosan oligosaccharides, respectively. Data are expressed as mean ± SEM (n = 12). One-way ANOVA with Dunnett's multiple comparisons test was performed.
(TIF)

**S2 Fig. Representative photos of an agar disk diffusion method.** Zone of inhibition (ZOI) of *S. aureus* was measured. Control represents supernatant obtained from bone cement specimens made solely from Copal® G+V (A, D). Ch 10% are supernatant obtained from specimens made of Copal® G+V mixed with 10% w/w chitosan (B, E). ChO 10% are supernatant obtained from specimens made of Copal® G+V mixed with 10% w/w chitosan oligosaccharides (C, F).
(TIF)

**S3 Fig. Representative photos of an agar disk diffusion method.** Zone of inhibition (ZOI) of MRSA was measured. Control represents supernatant obtained from bone cement specimens made solely from Copal® G+V (A, D). Ch 10% are supernatant obtained from specimens made of Copal® G+V mixed with 10% w/w chitosan (B, E). ChO 10% are supernatant obtained from specimens made of Copal® G+V mixed with 10% w/w chitosan oligosaccharides (C, F).
(TIF)

**S4 Fig. Relative cell viability (%) of Saos-2 cells after treatment with various concentration of SLS (25–100 μg/mL) for 24 h.** PBS-treated group was considered 100% cell viability. Data are expressed as mean ± SEM (n = 21–42).
(TIF)

**S1 Table. The minimal data set underlying the results.**
(PDF)

## Acknowledgments

The authors would like to thank Ms. Phatraya Srabua, Scientific and Technological Research Equipment Center (STREC), Chulalongkorn University, and Ms. Nanthawan Jinakul, Department of Microbiology, Faculty of Pharmacy, Mahidol University, for the SEM imaging and the antimicrobial assay, respectively. The authors thank Ms. Kanokwadee Sirithep, Hip Fracture Research Unit, Chulalongkorn University, and Ms. Awadsaya Pakdee, Faculty of Pharmacy, Mahidol University, for the lab technical support.

## Statement of ethics

All experiment sets in the study above were only related to *in vitro* experiments. There is no animals or human involved. As such, there is no need for any ethical declaration.

## Author Contributions

**Conceptualization:** Saran Tantavisut, Jiraporn Leanpolchareanchai, Amaraporn Wongrakpanich.

**Data curation:** Jiraporn Leanpolchareanchai, Amaraporn Wongrakpanich.

**Formal analysis:** Saran Tantavisut, Jiraporn Leanpolchareanchai, Amaraporn Wongrakpanich.

**Funding acquisition:** Saran Tantavisut, Amaraporn Wongrakpanich.

**Investigation:** Jiraporn Leanpolchareanchai, Amaraporn Wongrakpanich.

**Methodology:** Saran Tantavisut, Jiraporn Leanpolchareanchai, Amaraporn Wongrakpanich.

**Project administration:** Saran Tantavisut, Amaraporn Wongrakpanich.

**Supervision:** Amaraporn Wongrakpanich.

**Validation:** Saran Tantavisut, Jiraporn Leanpolchareanchai, Amaraporn Wongrakpanich.

**Visualization:** Jiraporn Leanpolchareanchai, Amaraporn Wongrakpanich.

**Writing – original draft:** Saran Tantavisut, Jiraporn Leanpolchareanchai, Amaraporn Wongrakpanich.

**Writing – review & editing:** Saran Tantavisut, Jiraporn Leanpolchareanchai, Amaraporn Wongrakpanich.

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
