## [Decision Letter · Decision Letter 0]

13 Jun 2022

PONE-D-22-14809Influence of chitosan and chitosan oligosaccharide on dual antibiotic-loaded bone cement: in vitro evaluationsPLOS ONE

Dear Dr. Wongrakpanich,

Thank you for submitting your manuscript to PLOS ONE. After careful consideration, we feel that it has merit but does not fully meet PLOS ONE’s publication criteria as it currently stands. Therefore, we invite you to submit a revised version of the manuscript that addresses the points raised during the review process.

We look forward to receiving your revised manuscript.

Kind regards,

Syed Mahmood

Academic Editor

PLOS ONE

Journal Requirements:

Reviewers' comments:

Reviewer's Responses to Questions

**Comments to the Author**

1. Is the manuscript technically sound, and do the data support the conclusions?

Reviewer #1: Yes

2. Has the statistical analysis been performed appropriately and rigorously? 

Reviewer #1: N/A

3. Have the authors made all data underlying the findings in their manuscript fully available?

Reviewer #1: Yes

4. Is the manuscript presented in an intelligible fashion and written in standard English?

Reviewer #1: No

5. Review Comments to the Author

Reviewer #1: 1. Language and grammar need to be improved. Many sentences are missing adjectives, and articles are not properly formatted, making them difficult to understand. Also, a few references cited in the text are incomplete.

2. The following sentences seem repetitive. Authors are kindly advised to improve the sentence structure to avoid confusion and improve readability.

Polymethylmethacrylate (PMMA) bone cement has many important roles in orthopaedics. It’s also played an imminent role in the treatment of orthopaedic infection.

In the current practice, ALBC is used as a prophylaxis to prevent prosthetic joint infections (PJIs) in total joint arthroplasty. Moreover, ALBC is also used to treat orthopaedic infections such as PJIs and chronic osteomyelitis.

3. “To prepare the Ch and the ChO mixed bone cement samples, different amounts of Ch and ChO were added to the polymethylmethacrylate matrix with three concentrations (1%, 5%, and 10%)”. Authors kindly specify the basis for the selection of the aforementioned conditions. Also, comment on why only three concentrations were taken.

4. As mentioned in the introduction section manual mixing has several limitations. Still, the technique followed for the present work is manual mixing. Authors kindly justify why this is selected.

5. For antibacterial activity why only S. aureus and MRSA were considered. Are they predominant causative agents for orthopaedic infection? Authors kindly comment.

6. Why is it preferable to use a pre-warmed PBS for drug elution assay?

6. PLOS authors have the option to publish the peer review history of their article (what does this mean?). If published, this will include your full peer review and any attached files.

Reviewer #1: **Yes: **Zeenat Iqbal

---

## [Author Response · Author response to Decision Letter 0]

20 Sep 2022

Reviewer #1

General comments: 

The submitted work is interesting as the author not only presented their work but has highlighted the limitations of the study. A few minor revisions were suggested. 

Specific comments/questions:

1. Language and grammar need to be improved. Many sentences are missing adjectives, and articles are not properly formatted, making them difficult to understand. Also, a few references cited in the text are incomplete.

Response: We have corrected spelling, grammar, and punctuation errors in the document. We also rewrote a few sentences. All changes were tracked. We also thoroughly reviewed the references.

2. The following sentences seem repetitive. Authors are kindly advised to improve the sentence structure to avoid confusion and improve readability. 

Polymethylmethacrylate (PMMA) bone cement has many important roles in orthopaedics. It’s also played an imminent role in the treatment of orthopaedic infection.

In the current practice, ALBC is used as a prophylaxis to prevent prosthetic joint infections (PJIs) in total joint arthroplasty. Moreover, ALBC is also used to treat orthopaedic infections such as PJIs and chronic osteomyelitis.

Response: We agreed with the reviewer and changed the sentences to improve readability (Location page 4 line 60-61 and Location page 4 line 63-66).

3. “To prepare the Ch and the ChO mixed bone cement samples, different amounts of Ch and ChO were added to the polymethylmethacrylate matrix with three concentrations (1%, 5%, and 10%)”. Authors kindly specify the basis for the selection of the aforementioned conditions. Also, comment on why only three concentrations were taken. 

Response: Since there have been no existing studies on chitosan oligosaccharides and PMMA bone cement, we chose these parameters based on earlier works of chitosan by Dunne et al. (1), Tunney et al. (2), and Tan et al. (3). Incorporating new material into bone cement may impair its mechanical properties. Dunne et al. used three different concentrations of chitosan (1, 3, and 5% w/w) in bone cement. The mechanical properties of bone cements mixed with chitosan were significantly reduced after the degradation period, especially in the 5%w/w chitosan group (1). Tunney et al. also found that the mechanical properties of bone cement containing chitosan (5% w/w) were significantly reduced. According to Tan et al., although mixing chitosan (in the form of quaternised chitosan) with bone cement at the concentration of 20% w/w would reduce the mechanical properties, the bone cement mixtures still passed the ISO 5883-2002 (3). Considering these data, three concentrations of Ch and ChO were chosen, covering the concentration ranging from 1 to 10% w/w. Low concentrations (1 and 5% w/w) were concentrations that were used previously. High concentration (10% w/w) was in the middle between studies conducted by Dunne et al., Tunney et al. and Tan et al.

4. As mentioned in the introduction section manual mixing has several limitations. Still, the technique followed for the present work is manual mixing. Authors kindly justify why this is selected.

Response: Despite the fact that various procedures, such as vacuum mixing, have been available for many years, surgeons continue to mix antibiotic-loaded bone cement (ALBC) by hand using only a mixing bowl and a spatula. In almost all cases, hand mixing is the primary technique used in operating rooms in Thailand. As a result, hand mixing was employed in this study to simulate real-life situations.

In terms of manual mixing powder, this study requires hand-mixing chitosan or chitosan oligosaccharides with PMMA powder before mixing the powder with liquid. This is due to the lack of commercially available ALBC containing chitosan or chitosan oligosaccharides.

5. For antibacterial activity why only S. aureus and MRSA were considered. Are they predominant causative agents for orthopaedic infection? Authors kindly comment.

Response: Yes, they are the most common pathogen in orthopaedic surgical site infections. According to a review by Saadatian-Elahi et al. (4), 20% of surgical site infections from studies found in the MEDLINE literature were primarily caused by S. aureus. In Germany, S. aureus caused approximately one-third of postoperative surgical site infections (5). 

With the widespread use of penicillin, S. aureus has developed antibiotic resistance. Latha et al. (6) found that among pathogens in orthopaedic infection, MRSA is the leading pathogen in India. MRSA is also a common pathogen associated with surgical site infection in Japan (7) and China (8). 

These details have been added to the manuscript to clarify the bacteria selection in the study (Location page 14, line 324-325).

6. Why is it preferable to use a pre-warmed PBS for drug elution assay? 

Response: The process of drug release varies depending on many factors, such as surface-area-to-volume ratio, drug/carrier interactions, and environmental factors (pH, temperature). Increasing the temperature of the system often increases drug release (9). For the drug elution assay, the temperature of the medium was set to body temperature (37°C). The PBS was removed and replaced with pre-warmed PBS (37°C) at a predetermined time. Although changing PBS at the volume of 1 mL in 30 mL was considered a small volume, replacing it with 1 mL of new PBS may cause the system temperature to change if the PBS was cold. To eliminate this potential variable, the PBS should be pre-warmed.

References

1. Dunne N, Buchanan F, Hill J, Newe C, Tunney M, Brady A, et al. In vitro testing of chitosan in gentamicin-loaded bone cement: no antimicrobial effect and reduced mechanical performance. Acta Orthop. 2008;79(6):851-60.

2. Tunney MM, Brady AJ, Buchanan F, Newe C, Dunne NJ. Incorporation of chitosan in acrylic bone cement: effect on antibiotic release, bacterial biofilm formation and mechanical properties. J Mater Sci Mater Med. 2008;19(4):1609-15.

3. Tan H, Ao H, Ma R, Tang T. Quaternised chitosan-loaded polymethylmethacrylate bone cement: Biomechanical and histological evaluations. J Orthop Translat. 2013;1(1):57-66.

4. Saadatian-Elahi M, Teyssou R, Vanhems P. Staphylococcus aureus, the major pathogen in orthopaedic and cardiac surgical site infections: a literature review. Int J Surg. 2008;6(3):238-45.

5. Hardtstock F, Heinrich K, Wilke T, Mueller S, Yu H. Burden of Staphylococcus aureus infections after orthopedic surgery in germany. BMC Infect Dis. 2020;20(1):233.

6. Latha T, Anil B, Manjunatha H, Chiranjay M, Elsa D, Baby N, et al. MRSA: The leading pathogen of orthopedic infection in a tertiary care hospital, South India. Afr Health Sci. 2019;19(1):1393-401.

7. Fukuda H, Sato D, Iwamoto T, Yamada K, Matsushita K. Healthcare resources attributable to methicillin-resistant Staphylococcus aureus orthopedic surgical site infections. Sci Rep. 2020;10(1):17059.

8. Yang Z, Wang J, Wang W, Zhang Y, Han L, Zhang Y, et al. Proportions of Staphylococcus aureus and methicillin-resistant Staphylococcus aureus in patients with surgical site infections in mainland China: a systematic review and meta-analysis. PLoS One. 2015;10(1):e0116079.

9. Paarakh MP, Jose PA, Setty C, Christoper G. Release kinetics–concepts and applications. Int J Pharm Res Technol. 2018;8(1):12-20.

---

## [Editor Report · Decision Letter 1]

11 Oct 2022

Influence of chitosan and chitosan oligosaccharide on dual antibiotic-loaded bone cement: in vitro evaluations

PONE-D-22-14809R1

Dear Dr. Wongrakpanich,

We’re pleased to inform you that your manuscript has been judged scientifically suitable for publication and will be formally accepted for publication once it meets all outstanding technical requirements.

Kind regards,

Syed Mahmood

Academic Editor

PLOS ONE
---

## [Editor Report · Acceptance letter]

18 Nov 2022

PONE-D-22-14809R1 

Influence of chitosan and chitosan oligosaccharide on dual antibiotic-loaded bone cement: *in vitro* evaluations 

Dear Dr. Wongrakpanich:

I'm pleased to inform you that your manuscript has been deemed suitable for publication in PLOS ONE. Congratulations! Your manuscript is now with our production department. 

Kind regards, 

on behalf of

Dr. Syed Mahmood 

Academic Editor

PLOS ONE